environmental science

disasters, climate change, heterogeneous response, monsoon, westerlies

**Author for correspondence:**
Zhibin Zhang
e-mail: zhangzb@ioz.ac.cn

†These authors contributed equally to this study.

# Climate change affected the spatio-temporal occurrence of disasters in China over the past five centuries

Chuan Yan[1,2,†], Huidong Tian[1,†], Xinru Wan[1], Jinxing He[1], Guoyu Ren[3,4], Ulf Büntgen[5,6,7], Nils Chr. Stenseth[8] and Zhibin Zhang[1]

[1]State Key Laboratory of Integrated Management on Pest Insects and Rodents in Agriculture, Institute of Zoology, Chinese Academy of Sciences, Beijing 100101, People's Republic of China
[2]State Key Laboratory of Grassland Agro-ecosystem, Institute of Innovation Ecology and College of Life Sciences, Lanzhou University, Lanzhou 730000, People's Republic of China
[3]Department of Atmospheric Science, School of Environmental Studies, China University of Geosciences, Hongshan District, Wuhan, People's Republic of China
[4]National Climate Center, China Meteorological Administration, Haidian District, Beijing, People's Republic of China
[5]Department of Geography, University of Cambridge, CB2 3EN Cambridge, UK
[6]Swiss Federal Research Institute WSL, Zürcherstrasse 111, 8903 Birmensdorf, Switzerland
[7]CzechGlobe, Global Change Research Institute CAS and Masaryk University, Kotlářská 2, 61137 Brno, Czech Republic
[8]Centre for Ecological and Evolutionary Synthesis, Department of Biosciences, University of Oslo, Blindern, 0316 Oslo, Norway

CY, 0000-0002-6669-432X; XW, 0000-0002-1580-9692; UB, 0000-0002-3821-0818; NCS, 0000-0002-1591-5399; ZZ, 0000-0003-2090-7999

Climate change may contribute to the spatio-temporal occurrence of disasters. Long-term studies of either homogeneous or heterogeneous responses of historical disasters to climate change are, however, limited by the quality and quantity of the available proxy data. Here we reconstruct spatio-temporal patterns of five types of disasters in China during the period AD 1368–1911. Our analyses of these time series reveal that warmer temperatures decreased the occurrence of disasters in the monsoon-affected parts of central-east China, but it increased the frequency and intensity of disasters along the boundary of arid and humid conditions in parts of southwest and northeast China, probably driven by the interplay among monsoon, westerlies, polar vortex and variation of temperature. Moreover, we show that drought and flood events had cascading effects on the occurrences of locust outbreaks, famine and human epidemics. Our findings suggest that climate can contribute

to the spatio-temporal occurrence of disasters, and therefore may contribute to an improvement of China's regional to national risk management of future climate and environmental change.

## 1. Introduction

Disasters, such as epidemics, pest outbreaks, drought and flood events, can all have huge impacts on both human societies and ecosystems. In historical and modern times, countless human lives have been lost due to various deadly climate-related disasters, such as infectious diseases and famine, in various regions of the world [1,2]. Frequent flooding and prolonged drought not only damage natural ecosystems [3,4] but also cause serious agricultural and societal problems [5]. Therefore, identifying the key driving forces behind such disaster occurrences in space and time is imperative in both natural and medical sciences.

Previous studies indicate that many disasters are driven by climate change. It is increasingly appreciated that global warming could contribute to many natural and biological calamities, yet easily attaining a consensus on this role has been elusive [6]. Regarding epidemics, short-term studies (usually less than 100 years) have shown that rising temperature is likely to increase the frequency of infectious diseases [7], while recent studies using long-term historical data have revealed that the prevalence of epidemics may be scale-dependent [8,9]. Many studies also suggest that global warming has triggered flood/drought disasters within the last century [10,11]. However, according to the records spanning a vast part of the last millennium or longer, long-term climate cooling was linked to an increase in the frequency of pest outbreaks [12] and other disasters [8,9].

Responses of disasters to climate change may be homogeneous at the continental scale because the impacts of climatic changes at that scale on disasters are often themselves continental. Yet disasters may exhibit a heterogeneous response to climate change across different regions due to climate's interaction with land and ocean characteristics. Much evidence confirms that the prevalence of various human epidemics was marked by spatial variation linked to host and environmental characteristics, such as in the case of the human plague [13] and malaria [14], among others. Another study revealed that pest outbreaks are spatially heterogeneous due to differing environmental factors and species interactions [15]. The IPCC AR5 report includes projections of the varied drought and flood risks faced in different regions under plausible climate warming scenarios [16]. Although the association of climate change with disasters is well-explored worldwide, studies on the homogeneous or heterogeneous response of disasters to spatial variation of climatic changes are scarce.

Different disasters might be closely associated in time and space, which is important for inferring causal relationships among disasters vis-à-vis their risk management. Climate change can cause compound and/or cascading disasters. The former refers to the multiple events occurring simultaneously or consecutively, and the latter stresses the sufficient forces or energy in the initial event to trigger the subsequent events [17]. For example, floods/droughts may lead to agricultural failures, and later, bouts of famine and social chaos [18], consequently promoting the emergence and spread of human epidemics [8,19]. In this way, a single disaster may be a risk-enhancing factor or it directly triggers another disaster's occurrence [5]. However, the spatial patterns of putative causal associations among disasters seem to be rarely investigated from a long-term perspective.

The associations between disasters and climatic changes are undoubtedly well investigated when relying upon short-term historical data of the modern age (usually less than 100 years). For instance, IPCC AR5 report expects an increase of average precipitation (from 1986–2005 to 2081–2100) in all parts of China under its global warming scenarios [16], implying a homogeneous association between temperature warming and drought/flood risks. Nevertheless, an examination of long-term spatio-temporal historical data of disasters is essential for revealing the region-specific response of disasters to climate change, but such an analysis has not been sufficiently performed so far. China has a long history of recording various disasters for over three millennia [8,20], providing us with the opportunity to study the homogeneous or heterogeneous response of disasters to long-term climate warming or cooling trends. The climate of China is controlled by several monsoon systems (i.e. East Asia summer/winter monsoon and South Asia summer monsoon) and the Westerlies [21], which might alter the spatial responses of disaster to climate change, particularly along the country's arid and monsoon zones where an unstable climate may be triggered.

Historical records have several limitations, including coarse spatial and temporal resolutions, unclear severity, unknown underlying mechanisms, etc. This makes it difficult to address the associations

between historical events and actual climate variability in full detail. For example, flood events might be caused by high precipitation extremes or rapid temperature increase following extensive winter snowfall. Nonetheless, this type of data still has advantages in revealing long-term effects of climate change. Hence in this study, we hypothesized that long-term climate warming or cooling would cause heterogeneous impacts on the occurrences of disasters due to complex interplay among land, ocean and monsoon systems. To address this, we reconstructed the spatio-temporal occurrence of five types of environmental disasters—human epidemic, famine, locust outbreak, drought and flooding—in China for over five centuries (spanning 1368–1911). Using this historical dataset, we focused on testing the specific hypothesis that large-scale temperature changes would show distinct associations with disasters in the arid and monsoon zones of China. To supplement that, we also examined the evidence for cascading effects of flood and drought on locust outbreaks, famines and human epidemics.

# 2. Material and methods

## 2.1. Disaster data

The data of China's disasters (human epidemics, locust outbreaks, famines, droughts and floods) were extracted from the book *A Compendium of Chinese Meteorological Records of the Last 3000 Years* [20], in which over 100 000 weather-related records were meticulously identified and catalogued. Each record was then carefully crosschecked for its consistency with other reports and organized by date (year) and location (mostly at the county or prefecture level). Most of the records came from standard histories and local gazetteers in the Ming-Qing period; these standard histories were compiled by governments while local gazetteers—including historical, geographical, economic, administrative and natural events—were compiled by local officials. The county-level locations of each record have already been documented and reported on [22], for which the geographical coordinates of the capitals of counties were used for further analyses (electronic supplementary material, figure S1). We only used data available from the Ming and Qing dynasties (AD 1368–1911) because there is little reliable data (in terms of both space and time data) before Ming and it is difficult to find. The records on human epidemics were unspecific and could have been the plague, malaria, cholera and other human infectious diseases. For details of data extraction and digitalization, see [8]. Our study only used the frequency data of disasters, without assigning severity levels to disaster events.

## 2.2. Climate variables

We focused on analysing the spatio-temporal associations between large-scale temperature changes and disasters' occurrences. To do this, one annually resolved (temp.M) and two decadally resolved (temp.L and temp.Y) reconstructed temperature series (for details, see electronic supplementary material, table S1) were used; temp.M and temp.L are two temperature proxies representing the Northern Hemisphere, and temp.Y represents the temperature of China. From our prior study [8], we knew that these temperature variables had close associations with the investigated disasters.

## 2.3. Event coincidence analysis

We carried out an event coincidence analysis (ECA), by following [5], to determine the association of different disasters, using the latter's presence versus absence of a given disaster type in grid cell at a specific time step. ECA is a Boolean method of (asymmetric) correlation for binary time series, and this method is especially powerful and useful for inferring the statistical interdependence of multiple disasters and extreme climate events [23]. First, the map of China was divided into a 100 × 100 km grid, to produce a binary time series of disasters in each grid cell. If the coordinate of a disaster event was within a grid cell, it is allocated to that cell. For a given year, disaster data were assigned a value of '1' for the presence of at least a disaster record, and conversely '0' for no disaster record in each grid cell. Considering the record frequency in space and time dimension, we used the 100 × 100 km grid cell so as to have more presence or absence data of different disasters. In each grid cell, the coincidence among the disaster time series was tested to detect their interdependence. By again following Schleussner *et al.* [5], we performed a 'risk enhancement test': this uses the precursor coincidence rate to indicate a risk-enhancing relationship, and applies a 'trigger test' based on the trigger coincidence to discern directional causation. The precursor coincidence rate denotes 'the fraction of A-type events that occurs

simultaneously with one B-type event', and the trigger coincidence rate denotes 'the fraction of B-type events that occurs simultaneously with one A-type event'. A high value for precursor coincidence may be interpreted as a high risk of A-type event enhanced by a B-type event, whereas a high value for trigger coincidence may be interpreted as a high probability of a B-type event leading to an A-type event. The statistical significance of a coincidence was tested by 1000 randomly shuffled time series, with the entire analysis done using the CoinCalc package [24] for the R software platform. In the ECA, we did not adjust the presence and absence data by recording effort in time because different disaster events in the same grid cell were equally affected by the recording effort.

## 2.4. Associations between climatic changes and disasters

We analysed the associations between time series of climate changes and disasters at both annual and decadal scales. To do this, a larger grid size was used for reconstructing the time series of counts of disasters. First, the map of China was divided into a $300 \times 300$ km grid to produce time series of the disasters using either their annual or decadal frequency data (i.e. number of records) for each grid cell. The larger size of grid cells helped to have a continuous time series by reducing too many zero data. The time series of disaster records could be biased, however, due to variation in the recording efforts; so we used the total number of meteorological-related records in each grid cell, which represented the recording effort, to adjust its time series by following the method of Tian *et al.* [8]. These reconstructed time series were adjusted by using the formula: $T = (\text{Gmax} + G)/(2G) \times \text{T.orig}$, where $T$ is the adjusted time series, T.orig is the original time series, $G$ is the smoothed meteorological record number from a generalized additive model (GAM), and Gmax is the maximum value of $G$. To remove long-term linear trends in time, the adjusted time series was log-transformed and further detrended, using linear regression. We only included those grid cells that had meteorological-related records for at least 15 years for annual data, or 10 decades for the decadal data.

Second, we analysed the associations of temperature proxies with the occurrences of disaster variables. In each grid cell, we tested for a positive or negative association of temperature proxies (linearly detrended) with each disaster variable by using Pearson's correlation. To identify any scale-dependent effects in the time domains [8], the annual/decadal disaster data, as well as corresponding temperature data, were decomposed into sub-centennial and multi-centennial scales series, by using a 100-year or 10-decade moving average filter (i.e. the multi-centennial series was the data smoothed by the 100-year or 10-decade moving average filter, and the sub-centennial series comprised the residuals of the 100-year or 10-decade moving average filter). The sign of each Pearson correlation coefficient represented a positive or negative response pattern in space, but its significance level may nonetheless be affected by temporal autocorrelation [25]. Finally, we compared our results for drought and flood occurrences with the trends in annual total precipitation in China from 1961–1965 to 1996–2000 [26].

# 3. Results

## 3.1. Associations between different disasters

For the period of 1368–1911, the records were mainly distributed in the central, southern, eastern and southwest parts of China (electronic supplementary material, figure S1). Coincidence analysis revealed significant associations among these disasters but large spatial heterogeneity in the magnitude of those associations (figure 1). Risk enhancement testing indicated there was significant precursor coincidence among these disasters over most areas with disaster records, except that locust outbreaks showed low coincidence with human epidemics and famine (figure 1*a*). The trigger test detected high trigger coincidences in terms of 'locust → famine', 'drought → famine', and 'drought → locust' (figure 1*b*). On average, the precursor coincidences were greater than the trigger coincidences, suggesting dominant risk-enhancing associations between disasters. These results showed clear bottom-up effects of climate change on disasters from drought and flood further acting on locust outbreaks, famines and human epidemics (figure 1*c*).

## 3.2. Impact of climatic changes

Considering the annual data, for all disasters, we found predominant negative associations with the annually resolved Northern Hemisphere temperature (temp.M) in central and east China. By

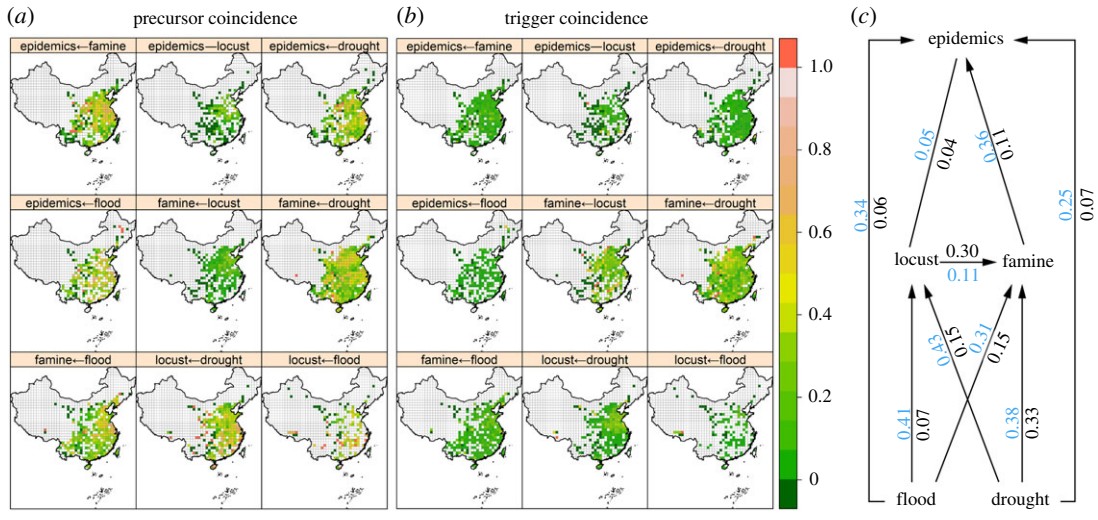

**Figure 1.** Spatial patterns of precursor (*a*) and trigger (*b*) coincidences among the five disaster type variables. The '←' presumes a causal relationship from the latter to former variable in *a* and *b*; in *c*, are shown the summarized average values of the precursor (blue) and trigger (black) coincidences. Empty grid cells indicate no data or non-significant coincidences.

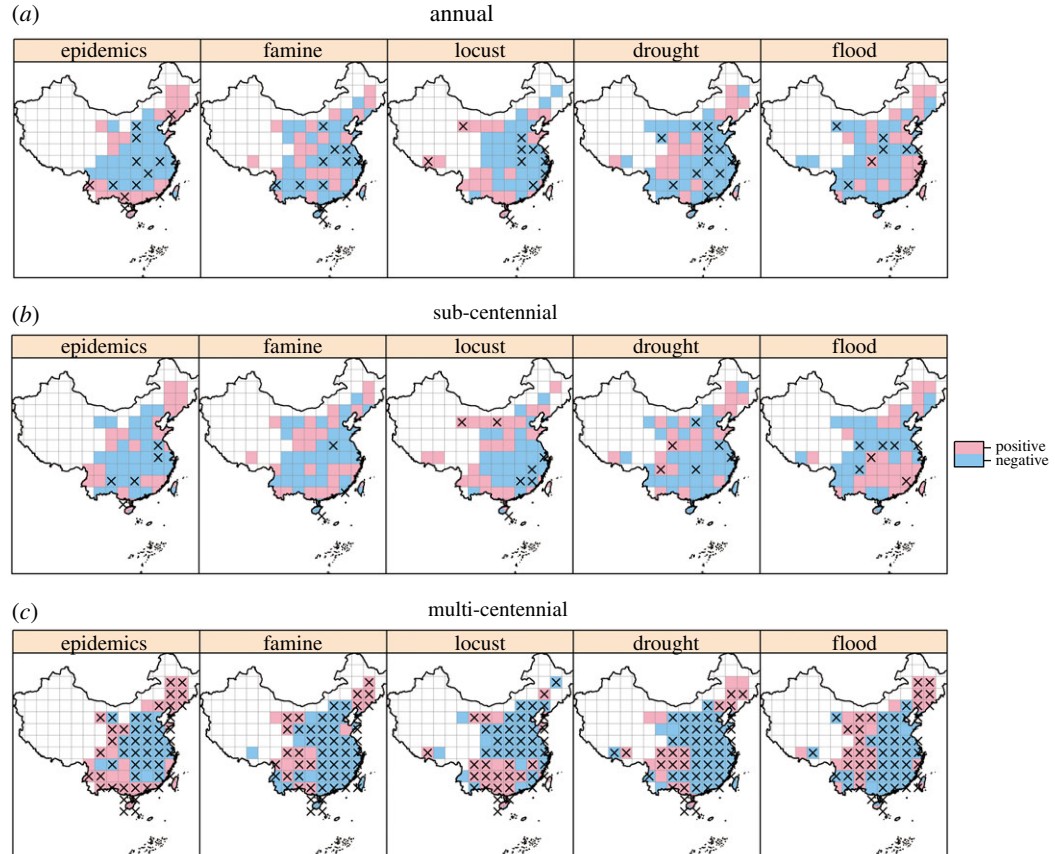

**Figure 2.** Spatial patterns of associations between the annual Northern Hemisphere temperature (temp.M) and the five disaster type variables at annual, sub-centennial and multi-centennial scales. The '×' indicates a grid cell of statistical significance (*p* < 0.05); empty grids have no available data.

contrast, those parts of northeast and southwest China near the boundary between the arid zone (west China) and monsoon zone (east China) featured more positive associations (figure 2*a*). However, temperature showed more positive associations with floods in many parts of south China, with human epidemics in coastal regions of south China and with more locusts in southwest China.

For the sub-centennial data, the results were similar to those for annual data except fewer significant negative associations were found, as more significant positive associations were evident with the annually resolved Northern Hemisphere temperature; but flood and famine showed positive associations with the annually resolved temperature in many parts of south China (figure 2b). Disasters across the boundary between arid and monsoon zones of China also showed more positive associations, whereas the opposite characterized flood (figure 2b).

For the multi-centennial data, each type of disaster showed predominant significant negative associations with annually resolved Northern Hemisphere temperature in central and east China, yet positive associations with temperature in parts of the country's northeast, south and southwest (figure 2c). Disasters across the boundary between arid and monsoon zones of China had more positive associations as well (figure 2c).

Similar results were obtained by using the decadally resolved temperature proxies (electronic supplementary material, figures S2–S4), but China's temperature (temp.Y) had more negative associations with all disasters than Northern Hemisphere temperature.

# 4. Discussion

Our study indicates that all five types of disasters showed positive responses to climate cooling in large parts of central and east China, yet negative responses to climate cooling (i.e. more disasters) did happen in a few other parts of China, particularly along the boundary of arid and monsoon zones during 1368–1911 timespan, supporting our key hypothesis. This was probably linked to temperature-driven alternation of the monsoons and Westerlies under the influence of climatic changes and China's unique geographical landscape (figure 3). Climatic changes had cascading associations among the occurrence of droughts, floods, locust outbreaks, famines and human epidemics. Additionally, our results on drought and flood were consistent with observed changes in precipitation from 1961–1965 to 1996–2000 in China (electronic supplementary material, figure S5).

Recent studies using historical data suggest that long-term temperature cooling may have promoted the long-term prevalence of multiple disasters in China over the past two millennia [8,9,12,19]. The associations between climate and natural or biological disasters often vary across spatial [6] or temporal [8] scales. Consistent with our previous finding [8], negative associations between temperature proxies and multiple disasters were predominant in both central and east China. Yet we also found positive associations in a few parts of southwest, south, north or northeast China, indicating that the response of certain disasters to climatic changes can be region-specific. Such a heterogeneous pattern of disasters' responses to a changing climate was more pronounced at the multi-centennial scale, and could involve distinct complex pathways. Based on the positive enhancing pathways found from drought and flooding to locust outbreaks, famines and epidemics, we believe drought and flood events might play a key role in structuring spatially heterogeneous responses of those disasters to climatic changes.

Risks of drought and flood are known to vary in different regions under different climate warming scenarios based on the data from contemporary climate records [27]. Our results revealed opposite responses of drought and flooding between central-east China and parts of southwest-northeast China: climate warming would reduce both disasters' occurrence in the former regions but increase it in the latter regions (figure 2). This interpretation supports previous projections made for northeast and southwest China, but not for the country's large central-eastern region, where majority of Chinese population live. The discrepancy could be due to the frequency components of data used. Much of the prior research has relied on short-term, contemporary data of climate mainly composed of sub-centennial data, whereas our climate data consists of both sub-centennial and multi-centennial components. For the sub-centennial data, we also uncovered many non-significant positive associations of drought or flood occurrence with a warming climate in central and east China (figure 2).

The distinct effects of temperature on flood and drought in arid and monsoon zones of China might be driven through monsoons and Westerlies. Plausible mechanisms stemming from our observations are illustrated in figure 3. In winter, the East Asian winter monsoon (EAWM) from Siberia and Mongolia plateau brings dry and cold wind to central and east China. It is well recognized that strong EAWNs spur to cold periods, causing droughts in north, central and east China [28]. In central and east China, summer precipitation is determined principally by the East Asian summer monsoon (EASM), which transports moisture from the Indian and Pacific Oceans to China. Two external forces affect EASM, i.e. thermal effect of the land–sea contrast and snow cover in Eurasia and on the Tibetan Plateau [29]. Cold temperatures over the landmass during summer may diminish the land–sea thermal gradient, possibly

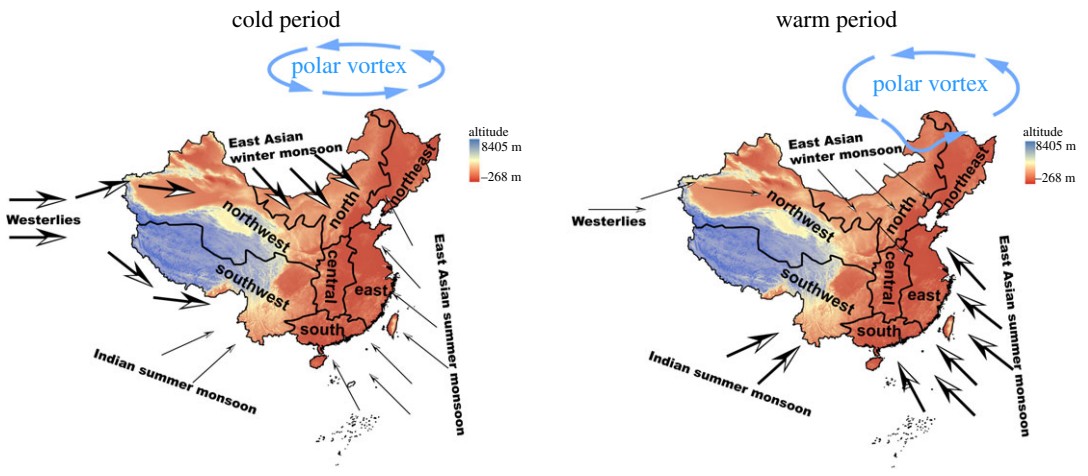

**Figure 3.** Illustrations of climate patterns controlled by monsoons in China, the Westerlies, and the polar vortex in cold and warm periods during the last millennium based on the literature (for details, see text). In cold period, Indian and East Asian summer monsoons are weak, Westerlies and East Asian winter monsoon are strong and Polar vortex is parked in polar regions; in warm period, Indian and East Asian summer monsoons are strong, Westerlies and East Asian winter monsoon are weak and Polar vortex expands southward. This contrasting pattern could be partially used to explain the observed associations between frequencies of five disasters and temperature in China.

leading to more droughts in the north and more frequent floods in the lower Yangtze River [30]. Further, winter snow over the Tibetan Plateau is positively correlated with rainfall in the following summer in the region of the Yangtze River basin and this relationship is mediated by EASM [29]; hence, enhanced snow cover during cold periods can also induce more flood occurrences [31]. In a cold phase, the high Equator-to-pole temperature gradient would cause more cyclones to form at mid-latitudes [32], which would increase the likelihood of flooding events in east China. The higher frequency of cyclones in this cold phase is probably linked to the unstable climate of cold periods. Our findings for central, east and parts of north China are consistent with those of previous work [8,12], but our results for China's other regions are not.

Notably, drought and flood events in historical records are not necessarily equal to the amount of precipitation. Floods may be caused by high precipitation extremes or seasonal melting of snowfall, but not necessarily by long-term trends in mean precipitation. Drought might not be only caused by the absence of precipitation, but also affected by accelerated evaporation due to high temperatures. Therefore, we need to be cautious in interpreting the association of drought and flood occurrences and temperature- or monsoon-driven precipitation.

It is widely accepted that the poles are warming faster than other parts of the Earth [33]. Consequently, global warming reduces the temperature gradient between tropical and polar regions, thus reducing the frequency of disasters by weakening extra-tropical storm activities in the monsoon regions of eastern China [34]. Indeed, we did find significant negative correlations between the decadal mean value and standard deviation (s.d.) of temp.M in decadal-moving windows (electronic supplementary material, figure S6). That the decadal s.d. of temp.M was positively—albeit not significantly in many grid cells—associated with all disaster types in most of China (figure 4) might also explain their negative responses to temperature changes in the largest regions (i.e. central and east China) which are heavily influenced by the East Asian Monsoon. These results suggest that unstable atmospheric circulation could increase the frequency of disasters. It is notable that the effects of mean value and standard deviation of temperature on disasters were roughly reversed in space (figure 2 versus figure 4). The similarity (anti-phase) between figures 2 and 4 is striking, indicating the observed patterns are robust. We speculate that climate cooling augmented the frequency of disasters in a large part of China, by not only affecting the monsoon and Westerlies or polar vortex but also promoting meridional atmospheric circulation, and vice versa.

Southwest China often experiences droughts in winter or spring, but floods in summer. More droughts are predicted to occur in southwest China against the background of global warming [35], in line with our results. Southwest China features a high elevation and subtropical monsoon climate, and a temperature rise there might increase evaporation, causing more severe droughts in winter and spring (dry season). One important circulation phenomenon affecting China's west is the Westerlies.

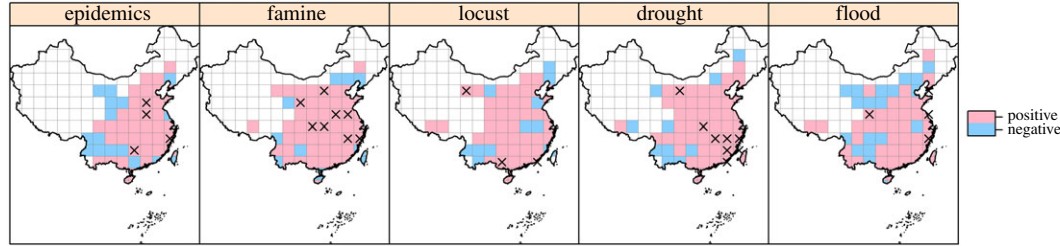

**Figure 4.** Associations of decadal standard deviation (s.d.) of temp.M with the decadal frequency of five disaster type variables. The '✕' indicates statistical significance ($p < 0.05$); empty grids lacked available data.

The Tibetan Plateau splits the Westerlies into north and south branches in winter; the south branch brings precipitation for southwest China [21]. Rising temperatures are likely to cause the northward shift of Westerlies, resulting in less precipitation and thus more droughts in southwest China; meanwhile, the latter's summer precipitation is also influenced by the Indian summer monsoon (ISM). Global warming intensifies the ISM through a strengthened land–ocean thermal gradient and increases moisture transport [36], consequently causing more summer flooding events in the rainy summer seasons in this area. Recent studies suggest climate warming would cause a range shift in tropical diseases towards the polar regions [37], supporting our results (figure 2).

Recent studies indicate that climate warming would enable dips of the Arctic vortex into high or middle latitude regions (e.g. northeast China), bringing forth more unusual climatic extreme events, like heavy snowfall, severe flooding or prolonged drought [38,39]. Thus, climate warming would leave north and northeast China more prone to severe flooding. Therefore, we speculate that Westerlies and polar vortex could explain the historical heterogeneous response to climatic changes in some parts of southwest and northeast China.

Although the data for west China were admittedly limited, our results did reveal opposite response of disasters in some grid cells between it and the central-east (figure 2). This may be explained by the different climate patterns of these two regions: the latter is mainly controlled by several monsoons, so climate warming would bring more moisture into these regions from tropical oceans, and vice versa. West China is mainly controlled by the Westerlies; hence, climate cooling would bring more moisture into west China from the Atlantic Ocean or Mediterranean Sea [21]. However, He *et al.* [26] found that recent climate warming was accompanied by an increase in precipitation in the arid zone of northwest and northeast China, suggesting that the East Asian and Indian summer monsoon currents might have penetrated to the region and brought more moisture and precipitation to the arid zone. Therefore, the precipitation of west China is probably controlled by both Westerlies and summer monsoon, depending on the degree of climate warming.

The responses at both sub- and multi-centennial levels were basically similar except that there were more significant associations between disaster occurrences and temperature at multi-centennial than at sub-centennial levels. Besides, floods showed contracting patterns between two scales: more flood events in many parts of south China occurred in short-term climate warming phase as compared with those in long-term cooling phase, supporting the observation by He *et al.* [26].

Many studies have shown that different disasters were often closely associated with each other, but their risk-enhancing and causal relationships are rarely investigated (but see Schleussner *et al.* [5]). Our prior work using time-series data provided evidence that drought and flood events increased epidemics' prevalence indirectly through locust outbreaks and bouts of famine at temporal scales [8], but evidence for this based on spatio-temporal analysis was still lacking. Here, our results not only support our prior observations, but they also revealed historical heterogeneous associations in space in the respective magnitude of risk-enhancing and causal relationships. Their spatial patterning was distinct between precursor and trigger coincidences. Notably, we found widespread higher precursor coincidences and lower trigger coincidences in the 'famine → epidemics', 'drought → epidemics', 'flood → epidemics', 'drought → locust', and 'flood → locust' cascades, suggesting drought and flood played significant roles in enhancing risk, instead of triggering human epidemics or locust (however, drought might trigger locust outbreaks in the northern China Plain). Many factors jointly determine the occurrence of human epidemics (e.g. host status, pathogen dynamics, transmission waves, sanitation) [40], famines (e.g. human population density, social structure) [41] and locust outbreaks (e.g. pest's habitat, demography) [42]. Accordingly, though drought and flood might not directly trigger the occurrences of these disasters, they do enhance their risks. The higher triggering coincidence along with

a lower precursor coincidence suggests locusts are more likely to trigger a famine rather than enhance it, because locust outbreaks could directly cause crop failure [43]. Drought was characterized by a high precursor and trigger coincidence with famine disasters, while flood only had a high precursor coincidence, suggesting drought, not flooding, played a more direct role in causing famine events in China. The distinct spatial pattern of risk-enhancing and triggering relationships of these disasters have important implications for risk management and prediction of disasters. For risk-enhancing factors, proactive strategies can be taken to reduce risks of relevant disasters; while for triggering factors, corresponding strategies ought to prevent the consequences of relevant disasters.

To sum up, as figure 3 shows, climate pattern in China is shaped by several monsoons, the Westerlies and the polar vortex in cold and warm periods, which generate homogeneous or heterogeneous spatial associations between temperature and disasters in either cold or warm periods. A cold climate would increase disasters by creating an unstable and dry climate in China's east. It may also promote a stronger EAWN and reduce land–ocean water circulation, bringing more drought events to north, central and east China. A cold climate would also lead to more flood events, by increasing cyclone formations due to higher Equator-to-pole temperature differences in central and east China, and also strengthen the Westerlies through a humid atmosphere, bringing more rainy weather (that is, fewer droughts) in parts of West China. An opposite pattern would appear in the warm phase, however, when China's northeast and southwest would show a different response to climate warming or cooling from its central and east regions. For parts of northeast China, a warm climate would cause the southward slippage of the arctic polar vortex that could promote extreme climatic-linked disasters, such as flooding in summer or droughts in spring. For parts of the country's southwest, climate warming might bring more droughts due to a northward shift of Westerlies, or more floods because of a strengthened ISM. Lastly, large temperature oscillations in the cold phase might also increase disasters' frequency.

Our results on the climate-related risks of disasters could have implications for the risk management of disasters. Under current global climate warming, there is likely to be a lower risk for disasters in the east monsoon regions of China (covering central, east, and parts of north and south China) where most citizens reside, but it may elevate the risk of disasters in the country's northeast and southwest and along the boundary of the arid and monsoon zones. Climate warming would increase human epidemics in islands or coastal parts of south China. However, we must be cautious about these predictions because they are based on data from the Little Ice Age (LIA) to early modern period. The relationship between disasters and climate changes during the Late Holocene may differ from that of modern time (Anthropocene). More research is needed to reveal the association of past and more recent disasters with ongoing climate warming for having a better prediction in this twenty-first century. The region-specific responses of disasters to climate change as revealed in our study merit investigation in other regions of East Asia where climate is governed by monsoon and Westerlies as well as the polar vortex.

Data accessibility. The data were provided as electronic supplementary material, files.
Authors' contributions. C.Y. and H.T. contributed equally to the manuscript; Z.Z. designed the study; C.Y. contributed to the data analysis; H.T. contributed towards the data collection and its curation; X.W., J.H., G.R., N.C.S. and U.B. all contributed towards the manuscript's entire preparation. All authors contributed towards writing and revising the manuscript.
Competing interests. We declare no competing interests.
Funding. This work was supported by the National Key R&D Program of China (grant no. 2017YFA0603304); the External Cooperation Program of BIC, of the Chinese Academy of Sciences (CAS) (grant no. 152111KYSB20150023); the Key International Cooperation Grant of National Natural Science Foundation of China (grant no. 31420103913); the ISZS/IUBS Program of Biological Consequence of Global Change (BCGC); and the National Key R&D Program of China (grant no. 2018YFA0605603). C.Y. was supported by both the Young Elite Scientist Sponsorship Program by CAST (grant no. 2017QNRC001) and the Youth Innovation Promotion Association, CAS (grant no. 2018108).
Acknowledgements. We are grateful to reviewers and editors for their valuable comments on this manuscript. We thank Dr Leif C. Stige from the University of Oslo, Norway, for his valuable comments on the manuscript.

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
