## [Peer Review File · Royal Society Open Science]

Review History

RSOS-200731.R0 (Original submission)

Review form: Reviewer 1

Is the manuscript scientifically sound in its present form?

Yes

Are the interpretations and conclusions justified by the results?

No

Is the language acceptable?

Yes

Do you have any ethical concerns with this paper?

No

Have you any concerns about statistical analyses in this paper?

No

Recommendation?

Major revision is needed (please make suggestions in comments)

Comments to the Author(s)

Comments on “Climate change has distinct cascading impacts on disasters’ occurrence in arid and monsoon zones of China over the past five centuries” by Yan et al.

This study depicted the relations among main disasters for China during 1368-1911 as well as explored the relations between disasters and climate changes. The results are interesting. Firstly, it is noted that many scales of grid cells were applied. There are 100- by 100-km and 300- by 300-km. It is unclear that why different size of grid cells were applied. Moreover, it is also unclear how to allocate the historical records into grid cell. The historical records are mostly referring to political regions, which are random polygon.

Secondly, it is unclear that how to treat the levels of disaster recorded by historical documents. The method of treating the disasters’ level would deeply affect the results for the impacts of climate changes.

Thirdly, this study explained the correlations between climate changes and disaster using existing studies. It is noted that climate changes exhibited a lot of time scales. The Chen et al (ref. 19) paid attentions on the millennium scale of climate variations during the Holocene while this study paid attentions on the decadal and annual variations during past millennium. The results from Chen et al (ref. 19) could not be directly used to explain this study. For instance, the winter monsoon variations may be reverse to summer monsoon variations at millennium scale but they are not at decadal and annual scale. Hence, it is needed to note the time scale for the Figure 3. Meanwhile, in the text, the climate variations at many scales, spanning seasonal scale to millennium scale are likely mixed together.

L301-302, “thus reducing frequency of disasters by creating a stable atmospheric circulation (zonal) in the monsoon regions of eastern China”. Where is this result from? Please insert the reference. Generally, following the IPCC, in the context of warming climate, the climate is unstable.

Review form: Reviewer 2**Is the manuscript scientifically sound in its present form?**

Yes

Are the interpretations and conclusions justified by the results?

Yes

Is the language acceptable?

Yes

Do you have any ethical concerns with this paper?

No

Have you any concerns about statistical analyses in this paper?

Yes

Recommendation?

Major revision is needed (please make suggestions in comments)

Comments to the Author(s)

The authors present a statistical analysis of the occurrence dates of different types of disasters along with long-term climate change for an about 550-years period in Chinese history (1368-1911). They report differential responses of different disaster types in different regions of China in terms of co-occurrence frequencies and long-term trends, thereby identifying links between different disaster types pointing to an elevated likelihood of cascading disasters across different “sectors”.

The manuscript is clearly written and presents the obtained results in a concise manner. I believe that it warrants publication, while a few points should be addressed before I can finally make a corresponding recommendation.

Specific comments:

- In the context of the current interest in compound events, it might be good to include a brief explanation on the use of the terms “compound” versus “cascading” in the context of the present work in the introduction of this manuscript.
- Climate change seems to be solely addressed in terms of long-term warming or cooling trends, while actual climate variability has far more facets including drying or precipitation increases that seem to be overlooked in this work, possibly because they might exhibit less clear or more complex patterns in time and space. One of their disaster types (flood) could be considered closely associated with high precipitation extremes, yet not necessarily long-term trends in mean precipitation, and might also be heavily affected by the season (e.g. flooding due to extensive winter snowfall followed by rapid temperature increase). On the other hand, another type (drought) might be solely related to absence of precipitation, but also affected by high temperatures (depending on which notion of drought is considered, which is possibly hard to identify based on the underlying historical documents). I understand that not all the aforementioned aspects can be addressed in full detail in the context of the present study, but I would like to invite the authors to briefly discuss their linkages and thereby outline the actual implications, hypotheses and limitations of the present work in a more transparent way in the Introduction section.
- Related to the previous point, some more words on the possible reporting bias in data obtained from historical documents might be in order. Especially in terms of temporal trends (in event frequencies etc.), this might be relevant. Commonly, one might expect to find clusters of events towards the end of historical event series. If this happens, also co-occurrence frequencies (respectively, the p-values of the performed statistical tests) will be necessarily elevated. The shuffling based test for significance of event coincidence rates ignores such important serial dependencies within the data and thereby may lead to false positive results. While I do not claim that this is necessarily the case in the present study, I ask the authors to clarify this aspect.
- Regarding the statistical terminology, starting with Donges et al. (2016) (ref. 21), it has become common convention to term the employed method “event coincidence analysis” (ECA) in order to make this approach recognizable in comparison with the use of the term “coincidence analysis” in other contexts. In l. 127, it is stated that ECA is a causal inference method - in statistics, the term causal inference is however used in a somewhat different way commonly implying “predictive causality” rather than the “probability of (time-lagged) co-occurrences” provided by ECA, which has more a flavor of an (asymmetric) correlation-like measure for the statistical association between event sequences.
- I acknowledge that the authors consider a separation of the considered time series into low-frequency (moving-average) and high-frequency (residual) variations for the correlation analysis. As per their choice of the averaging window of 100 years, wouldn't it make more sense to speak about (multi)-centennial and sub-centennial variability? Particularly, I find the use of the term “all frequency” a bit odd.
- In ll. 226-227, when speaking of “negative responses”, do you mean “more disasters” or a positive association with temperature?

Technical suggestions:

- L.43: As this is an enumeration of components of the Earth system which together make up climate variability, I think it would be more reasonable to replace “climate” by “atmosphere”.
- LL.53-54: “millennium records” might be better phrased as “records spanning a vast part of the last millennium or longer” or something similar
- L.56: I suggest replacing the term “large” by something more specific.
- L.63: “spatially heterogeneous”
- L.82: “...an analysis has not been often performed so far.”
- LL.158-159: remove “against to year”, it should be clear from the context that “detrending” refers to removing a trend in time
- L.163: “grid cell”
- LL.239-242: As the paper discusses climate change and associated impacts during the last millennium, I am wondering if the remarks in the discussion regarding early and mid-Holocene monsoon variability are relevant in this context.
- L.269: Do you really refer to “predictions” (in the sense of statistical models) or rather “projections” (in terms of results of climate models with prescribed “boundary conditions”)?
- L.270: “80% of the Chinese population”
- LL.277-279: I think that the caption should provide more explanation on what is shown in the figures.
- L.281: “temperature-driven monsoons” – are there monsoons where temperature does not play the role of a driver? Similar applies in L.340 to the westerlies.
- LL.295, 300: Instead of “differential”, one might more commonly use the term “gradient”.
- LL.305, 319 and caption of Fig. S6: “standard deviation” (not “standardized”)
- L.323: I do not see where the authors have attempted any prediction in the present work.
- LL.361-363: Disentangling direct from indirect effects would require conditional versions of the presented analysis, e.g. conditional ECA (Siegmund et al., *Front. Plant Sci.*, 2016).
- L.368: replace “coupled to” by “along with” or something similar
- Please provide complete references for refs. 10, 27 and 33.

Decision letter (RSOS-200731.R0)

Dear Dr Zhang

The Editors assigned to your paper RSOS-200731 "Climate change has distinct cascading impacts on disasters' occurrence in arid and monsoon zones of China over the past five centuries" have now received comments from reviewers and would like you to revise the paper in accordance with the reviewer comments and any comments from the Editors. Please note this decision does not guarantee eventual acceptance.

We do not generally allow multiple rounds of revision so we urge you to make every effort to fully address all of the comments at this stage. The comments primarily address the clarity of the

text (e.g. in technical details, or to acknowledge that some conclusions drawn may require further critical examination).

If deemed necessary by the Editors, your manuscript will be sent back to one or more of the original reviewers for assessment. If the original reviewers are not available, we may invite new reviewers.

Please submit your revised manuscript and required files (see below) no later than 21 days from today's (ie 20-Oct-2020) date. Note: the ScholarOne system will 'lock' if submission of the revision is attempted 21 or more days after the deadline. If you do not think you will be able to meet this deadline please contact the editorial office immediately.

Best regards,

on behalf of Professor Len Thomas (Associate Editor) and Peter Haynes (Subject Editor)
openscience@royalsociety.org

Associate Editor Comments to Author (Professor Len Thomas):

We have now received two reviews of your manuscript, and both reviewers agree that your work is potentially worthy of publication. However, both made a number of points that need to be addressed before acceptance. I do not believe any of these points require a re-analysis, but substantial textual changes are suggested and so I am recommending major revision. In making changes, please make sure you carefully consider each reviewer comment and let us know in your cover letter how you have dealt with it; if you disagree strongly with a point they make, please explain your perspective in your cover letter. I look forward to seeing your revised manuscript.

Reviewer comments to Author:

Reviewer: 1
Comments to the Author(s)

Comments on "Climate change has distinct cascading impacts on disasters' occurrence in arid and monsoon zones of China over the past five centuries" by Yan et al.

This study depicted the relations among main disasters for China during 1368-1911 as well as explored the relations between disasters and climate changes. The results are interesting.

Firstly, it is noted that many scales of grid cells were applied. There are 100- by 100-km and 300- by 300-km. It is unclear that why different size of grid cells were applied. Moreover, it is also unclear how to allocate the historical records into grid cell. The historical records are mostly referring to political regions, which are random polygon.

Secondly, it is unclear that how to treat the levels of disaster recorded by historical documents. The method of treating the disasters' level would deeply affect the results for the impacts of climate changes.

Thirdly, this study explained the correlations between climate changes and disaster using existing studies. It is noted that climate changes exhibited a lot of time scales. The Chen et al (ref. 19) paid attentions on the millennium scale of climate variations during the Holocene while this study paid attentions on the decadal and annual variations during past millennium. The results from Chen et al (ref. 19) could not be directly used to explain this study. For instance, the winter monsoon variations may be reverse to summer monsoon variations at millennium scale but they are not at decadal and annual scale. Hence, it is needed to note the time scale for the Figure 3. Meanwhile, in the text, the climate variations at many scales, spanning seasonal scale to millennium scale are likely mixed together.

L301-302, "thus reducing frequency of disasters by creating a stable atmospheric circulation (zonal) in the monsoon regions of eastern China". Where is this result from? Please insert the reference. Generally, following the IPCC, in the context of warming climate, the climate is unstable.

Reviewer: 2

Comments to the Author(s)

The authors present a statistical analysis of the occurrence dates of different types of disasters along with long-term climate change for an about 550-years period in Chinese history (1368-1911). They report differential responses of different disaster types in different regions of China in terms of co-occurrence frequencies and long-term trends, thereby identifying links between different disaster types pointing to an elevated likelihood of cascading disasters across different "sectors".

The manuscript is clearly written and presents the obtained results in a concise manner. I believe that it warrants publication, while a few points should be addressed before I can finally make a corresponding recommendation.

Specific comments:

- In the context of the current interest in compound events, it might be good to include a brief explanation on the use of the terms "compound" versus "cascading" in the context of the present work in the introduction of this manuscript.
- Climate change seems to be solely addressed in terms of long-term warming or cooling trends, while actual climate variability has far more facets including drying or precipitation increases that seem to be overlooked in this work, possibly because they might exhibit less clear or more complex patterns in time and space. One of their disaster types (flood) could be considered closely associated with high precipitation extremes, yet not necessarily long-term trends in mean precipitation, and might also be heavily affected by the season (e.g. flooding due to extensive winter snowfall followed by rapid temperature increase). On the other hand, another type (drought) might be solely related to absence of precipitation, but also affected by high temperatures (depending on which notion of drought is considered, which is possibly hard to identify based on the underlying historical documents). I understand that not all the aforementioned aspects can be addressed in full detail in the context of the present study, but I would like to invite the authors to briefly discuss their linkages and thereby outline the actual implications, hypotheses and limitations of the present work in a more transparent way in the Introduction section.

- Related to the previous point, some more words on the possible reporting bias in data obtained from historical documents might be in order. Especially in terms of temporal trends (in event frequencies etc.), this might be relevant. Commonly, one might expect to find clusters of events towards the end of historical event series. If this happens, also co-occurrence frequencies (respectively, the p-values of the performed statistical tests) will be necessarily elevated. The shuffling based test for significance of event coincidence rates ignores such important serial dependencies within the data and thereby may lead to false positive results. While I do not claim that this is necessarily the case in the present study, I ask the authors to clarify this aspect.
- Regarding the statistical terminology, starting with Donges et al. (2016) (ref. 21), it has become common convention to term the employed method “event coincidence analysis” (ECA) in order to make this approach recognizable in comparison with the use of the term “coincidence analysis” in other contexts. In l. 127, it is stated that ECA is a causal inference method – in statistics, the term causal inference is however used in a somewhat different way commonly implying “predictive causality” rather than the “probability of (time-lagged) co-occurrences” provided by ECA, which has more a flavor of an (asymmetric) correlation-like measure for the statistical association between event sequences.
- I acknowledge that the authors consider a separation of the considered time series into low-frequency (moving-average) and high-frequency (residual) variations for the correlation analysis. As per their choice of the averaging window of 100 years, wouldn't it make more sense to speak about (multi)-centennial and sub-centennial variability? Particularly, I find the use of the term “all frequency” a bit odd.
- In ll. 226-227, when speaking of “negative responses”, do you mean “more disasters” or a positive association with temperature?

Technical suggestions:

- L.43: As this is an enumeration of components of the Earth system which together make up climate variability, I think it would be more reasonable to replace “climate” by “atmosphere”.
- L.53-54: “millennium records” might be better phrased as “records spanning a vast part of the last millennium or longer” or something similar
- L.56: I suggest replacing the term “large” by something more specific.
- L.63: “spatially heterogeneous”
- L.82: “...an analysis has not been often performed so far.”
- L.158-159: remove “against to year”, it should be clear from the context that “detrending” refers to removing a trend in time
- L.163: “grid cell”
- L.239-242: As the paper discusses climate change and associated impacts during the last millennium, I am wondering if the remarks in the discussion regarding early and mid-Holocene monsoon variability are relevant in this context.
- L.269: Do you really refer to “predictions” (in the sense of statistical models) or rather “projections” (in terms of results of climate models with prescribed “boundary conditions”)?
- L.270: “80% of the Chinese population”
- L.277-279: I think that the caption should provide more explanation on what is shown in the figures.
- L.281: “temperature-driven monsoons” – are there monsoons where temperature does not play the role of a driver? Similar applies in l.340 to the westerlies.
- L.295, 300: Instead of “differential”, one might more commonly use the term “gradient”.
- L.305, 319 and caption of Fig. S6: “standard deviation” (not “standardized”)
- L.323: I do not see where the authors have attempted any prediction in the present work.
- L.361-363: Disentangling direct from indirect effects would require conditional versions of the presented analysis, e.g. conditional ECA (Siegmund et al., Front. Plant Sci., 2016).
- L.368: replace “coupled to” by “along with” or something similar
- Please provide complete references for refs. 10, 27 and 33.

===PREPARING YOUR MANUSCRIPT===

===PREPARING YOUR REVISION IN SCHOLARONE===

- An individual file of each figure (EPS or print-quality PDF preferred [either format should be produced directly from original creation package], or original software format).
 - An editable file of each table (.doc, .docx, .xls, .xlsx, or .csv).
 - An editable file of all figure and table captions.
- Note: you may upload the figure, table, and caption files in a single Zip folder.
- Any electronic supplementary material (ESM).
 - If you are requesting a discretionary waiver for the article processing charge, the waiver form must be included at this step.
 - If you are providing image files for potential cover images, please upload these at this step, and inform the editorial office you have done so. You must hold the copyright to any image provided.
 - A copy of your point-by-point response to referees and Editors. This will expedite the preparation of your proof.

- Ensure that your data access statement meets the requirements at <https://royalsociety.org/journals/authors/author-guidelines/#data>. You should ensure that you cite the dataset in your reference list. If you have deposited data etc in the Dryad repository, please include both the 'For publication' link and 'For review' link at this stage.
- If you are requesting an article processing charge waiver, you must select the relevant waiver option (if requesting a discretionary waiver, the form should have been uploaded at Step 3 'File upload' above).
- If you have uploaded ESM files, please ensure you follow the guidance at <https://royalsociety.org/journals/authors/author-guidelines/#supplementary-material> to include a suitable title and informative caption. An example of appropriate titling and captioning may be found at https://figshare.com/articles/Table_S2_from_Is_there_a_trade-off_between_peak_performance_and_performance_breadth_across_temperatures_for_aerobic_scope_in_teleost_fishes_/3843624.

Author's Response to Decision Letter for (RSOS-200731.R0)

See Appendix A.

RSOS-200731.R1 (Revision)

Review form: Reviewer 1

Is the manuscript scientifically sound in its present form?

Yes

Are the interpretations and conclusions justified by the results?

Yes

Is the language acceptable?

Yes

Do you have any ethical concerns with this paper?

No

Have you any concerns about statistical analyses in this paper?

No

Recommendation?

Accept as is

Comments to the Author(s)

I have seen that the manuscript has been improved largely. It may be acceptable. I recommend to accept it.

Decision letter (RSOS-200731.R1)

Dear Dr Zhang,

It is a pleasure to accept your manuscript entitled "Climate change affected the spatiotemporal occurrence of disasters in China over the past five centuries" in its current form for publication in Royal Society Open Science. The comments of the reviewer(s) who reviewed your manuscript are included at the foot of this letter.

on behalf of Professor Len Thomas (Associate Editor) and Peter Haynes (Subject Editor)
openscience@royalsociety.org

Associate Editor Comments to Author (Professor Len Thomas):

Comments to the Author:

Thank-you for making the changes requested. Both I an a reviewer now agree the paper is ready for publication, and I am recommending acceptance.

Reviewer comments to Author:

Reviewer: 1

Comments to the Author(s)

I have seen that the manuscript has been improved largely. It may be acceptable. I recommend to accept it.

Appendix A

Cover letter

Dear Editor

Thanks for your decision letter. We are grateful to you for considering our manuscript after further revisions.

We have revised the manuscript based on the comments by you and the two reviewers.

Point to point response to reviewer's comments is attached with the cover letter.

Please let us know if you need us to make further clarification or revisions.

Regards

Zhibin Zhang

Appendix: Point to point responses to the reviewer's comments

Associate Editor Comments to Author (Professor Len Thomas):

We have now received two reviews of your manuscript, and both reviewers agree that your work is potentially worthy of publication. However, both made a number of points that need to be addressed before acceptance. I do not believe any of these points require a re-analysis, but substantial textual changes are suggested and so I am recommending major revision. In making changes, please make sure you carefully consider each reviewer comment and let us know in your cover letter how you have dealt with it; if you disagree strongly with a point they make, please explain your perspective in

your cover letter. I look forward to seeing your revised manuscript.

Response: Thank you very much for giving the opportunity to revise the manuscript.

We have revised the manuscript by following the comments from the editors and reviewers. Please see details below.

Reviewer comments to Author:

Reviewer: 1

Comments to the Author(s)

Comments on "Climate change has distinct cascading impacts on disasters' occurrence in arid and monsoon zones of China over the past five centuries" by Yan et al.

This study depicted the relations among main disasters for China during 1368-1911 as well as explored the relations between disasters and climate changes. The results are interesting.

Firstly, it is noted that many scales of grid cells were applied. There are 100- by 100-km and 300- by 300-km. It is unclear that why different size of grid cells were applied.

Moreover, it is also unclear how to allocate the historical records into grid cell. The historical records are mostly referring to political regions, which are random polygon.

Response: (1) The different scales were used for different analyses. The Event coincidence analysis needs only presence/absence data, so a smaller scale is used to represent as many occurrence data as possible. However, in the correlation analyses on associations between time series of climatic changes and disasters, we used a larger scale to have a large sample size of occurrence record in each cell for correlation analysis. (2) We have assigned the geographic coordinate of the capitals of political regions to disaster events. If the coordinate of a disaster event was within a grid cell, it is allocated to that cell. We have added the above more detailed information to clarify the methods. (L 119,142-144,166-168)

Secondly, it is unclear that how to treat the levels of disaster recorded by historical documents. The method of treating the disasters' level would deeply affect the results for the impacts of climate changes.

Response: In this work, we did not assign severity levels for disaster events because we focused on the frequencies of disasters. We used the original record data (i.e. presence or absence of a disaster event in a grid cell). We now have clarified this in the methods. (L 126-127)

Thirdly, this study explained the correlations between climate changes and disaster using existing studies. It is noted that climate changes exhibited a lot of time scales. The Chen et al (ref. 19) paid attentions on the millennium scale of climate variations during the Holocene while this study paid attentions on the decadal and annual variations during

past millennium. The results from Chen et al (ref. 19) could not be directly used to explain this study. For instance, the winter monsoon variations may be reverse to summer monsoon variations at millennium scale but they are not at decadal and annual scale. Hence, it is needed to note the time scale for the Figure 3. Meanwhile, in the text, the climate variations at many scales, spanning seasonal scale to millennium scale are likely mixed together.

Response: We deleted the contents about comparison with the results by Chen et al, and focused on the discussion on the effects of climate on monsoon and wetness at the annual and centennial scales.

L301-302, “thus reducing frequency of disasters by creating a stable atmospheric circulation (zonal) in the monsoon regions of eastern China”. Where is this result from? Please insert the reference. Generally, following the IPCC, in the context of warming climate, the climate is unstable.

Response: We delete this sentence.

Reviewer: 2

Comments to the Author(s)

The authors present a statistical analysis of the occurrence dates of different types of disasters along with long-term climate change for an about 550-years period in Chinese

history (1368-1911). They report differential responses of different disaster types in different regions of China in terms of co-occurrence frequencies and long-term trends, thereby identifying links between different disaster types pointing to an elevated likelihood of cascading disasters across different “sectors”. The manuscript is clearly written and presents the obtained results in a concise manner. I believe that it warrants publication, while a few points should be addressed before I can finally make a corresponding recommendation.

Response: We have revised the manuscript following your comments.

Specific comments:

- In the context of the current interest in compound events, it might be good to include a brief explanation on the use of the terms “compound” versus “cascading” in the context of the present work in the introduction of this manuscript.

Response: We now added an explanation on the “compound” versus “cascading” disasters in the Introduction.(L 70-73)

- Climate change seems to be solely addressed in terms of long-term warming or cooling trends, while actual climate variability has far more facets including drying or precipitation increases that seem to be overlooked in this work, possibly because they might exhibit less clear or more complex patterns in time and space. One of their disaster types (flood) could be considered closely associated with high precipitation extremes, yet not necessarily long-term trends in mean precipitation, and might also be heavily affected by the season (e.g. flooding due to extensive winter snowfall followed by rapid temperature

increase). On the other hand, another type (drought) might be solely related to absence of precipitation, but also affected by high temperatures (depending on which notion of drought is considered, which is possibly hard to identify based on the underlying historical documents). I understand that not all the aforementioned aspects can be addressed in full detail in the context of the present study, but I would like to invite the authors to briefly discuss their linkages and thereby outline the actual implications, hypotheses and limitations of the present work in a more transparent way in the Introduction section.

Response: We agree with this argument. Indeed, precipitation has complex association with floods which is more related to river regions. We added a paragraph in the discussion section. We also highlighted the contrasting patterns between two scales for floods and droughts (L287-294 , 337-348).

- Related to the previous point, some more words on the possible reporting bias in data obtained from historical documents might be in order. Especially in terms of temporal trends (in event frequencies etc.), this might be relevant. Commonly, one might expect to find clusters of events towards the end of historical event series. If this happens, also co-occurrence frequencies (respectively, the p-values of the performed statistical tests) will be necessarily elevated. The shuffling based test for significance of event coincidence rates ignores such important serial dependencies within the data and thereby may lead to

false positive results. While I do not claim that this is necessarily the case in the present study, I ask the authors to clarify this aspect.

Response: Indeed, there are more records in recent times for the whole China. In the event coincidence analysis, the influence of recording effort was same for different disasters' occurrence in the same grid cell, thus did not affect the significant dependence test. For correlation analysis, recording trend will affect the results. We corrected our data by using recording effort, and by using linear detrending filter. We have clarified this issue in the text (L 159-161)

- Regarding the statistical terminology, starting with Donges et al. (2016) (ref. 21), it has become common convention to term the employed method “event coincidence analysis” (ECA) in order to make this approach recognizable in comparison with the use of the term “coincidence analysis” in other contexts. In l. 127, it is stated that ECA is a causal inference method – in statistics, the term causal inference is however used in a somewhat different way commonly implying “predictive causality” rather than the “probability of (time-lagged) co-occurrences” provided by ECA, which has more a flavor of an (asymmetric) correlation-like measure for the statistical association between event sequences.

Response: Thanks for reminding us this difference. We now have clarified this in the methods section. (L136)

- I acknowledge that the authors consider a separation of the considered time series into low-frequency (moving-average) and high-frequency (residual) variations for the correlation analysis. As per their choice of the averaging window of 100 years, wouldn't it make more sense to speak about (multi)-centennial and sub-centennial variability?

Particularly, I find the use of the term "all frequency" a bit odd.

Response: We replaced the All, Low-, and High-frequency" with "Annual, sub-centennial and multi-centennial" as suggested.

- In ll. 226-227, when speaking of "negative responses", do you mean "more disasters" or a positive association with temperature?

Response: we have clarified this (L 234). It meant "more disasters in response to climate cooling".

Technical suggestions:

- L.34: As this is an enumeration of components of the Earth system which together make up climate variability, I think it would be more reasonable to replace "climate" by "atmosphere".

Response: This sentence is already deleted.

- Ll.53-54: "millennium records" might be better phrased as "records spanning a vast part of the last millennium or longer" or something similar

Response: corrected as suggested. L55

- L.56: I suggest replacing the term “large” by something more specific.

Response: modified as “ at the continental scale”. L57

- L.63: “spatially heterogeneous”

Response: corrected as suggested. L64

- L.82: “...an analysis has not been often performed so far.”

Response: corrected as suggested. L85-86

- LI.158-159: remove “against to year”, it should be clear from the context that “detrending” refers to removing a trend in time

Response: corrected as suggested.

- L.163: “grid cell”

Response: corrected as suggested.

- LI.239-242: As the paper discusses climate change and associated impacts during the last millennium, I am wondering if the remarks in the discussion regarding early and mid-Holocene monsoon variability are relevant in this context.

Response: we deleted this part.

- L.269: Do you really refer to “predictions” (in the sense of statistical models) or rather “projections” (in terms of results of climate models with prescribed “boundary conditions”)?

Response: corrected as “projections”. L260

- L.270: “80% of the Chinese population”

Response: corrected as suggested. L261

- LI.277-279: I think that the caption should provide more explanation on what is shown in the figures.

Response: we have added more content to explain the figure.

- L.281: "temperature-driven monsoons" – are there monsoons where temperature does not play the role of a driver? Similar applies in l.340 to the westerlies.

Response: we have deleted temperature-driven.

- LI.295, 300: Instead of "differential", one might more commonly use the term "gradient".

Response: corrected as suggested throughout the text.

- LI.305, 319 and caption of Fig. S6: "standard deviation" (not "standardized")

Response: corrected.

- L.323: I do not see where the authors have attempted any prediction in the present work.

Response: corrected as "our results". L312

- LI.361-363: Disentangling direct from indirect effects would require conditional versions of the presented analysis, e.g. conditional ECA (Siegmond et al., Front. Plant Sci., 2016).

Response: we deleted the two words.

- L.368: replace "coupled to" by "along with" or something similar.

Response: corrected as suggested. L367

- Please provide complete references for refs. 10, 27 and 33.

Response: corrected